# Nitrogen Addition Affects Interannual Variation in Seed Production in a Tibetan Perennial Herb

**DOI:** 10.3390/biology12081132

**Published:** 2023-08-14

**Authors:** Yuanxin Lou, Ruolan Wang, Peiyue Che, Chuan Zhao, Yali Chen, Yangheshan Yang, Junpeng Mu

**Affiliations:** 1Ecological Security and Protection Key Laboratory of Sichuan Province, Mianyang Normal University, Mianyang 621000, China; 18623782240@163.com (Y.L.); wangruolan2023@126.com (R.W.); a28589924@163.com (P.C.); chenyali5549@163.com (Y.C.); 2Key Laboratory of Mountain Ecological Restoration and Bioresource Utilization, Chengdu Institute of Biology, Chinese Academy of Sciences, Chengdu 610041, China; zhaochuan@cib.ac.cn; 3School of Ecological and Environmental Sciences, East China Normal University, Shanghai 200241, China; yangheshanyang@gmail.com

**Keywords:** reproductive success, biodiversity, plant-pollinator interactions, nitrogen applies, alpine meadow

## Abstract

**Simple Summary:**

There are various factors linked to global change that possess the capacity to alter the interannual variability in seed production. Our current knowledge regarding the impact of nitrogen availability on the year-to-year fluctuations in seed production patterns of perennial plants is limited. A multiyear field experiment was undertaken to examine the impact of nitrogen addition on the interannual seed production patterns of perennial plants. The introduction of element N had a significant impact on the preservation of aboveground biomass in plants, as well as the stability of flower traits. Consequently, this factor had an impact on the frequency of pollinator preference and the annual seed output. The findings of our study indicate that increasing the nitrogen content in the soil has the potential to alter the natural fluctuations in seed production that occur over different years. The results of this study possess the capacity to improve our understanding of the effects of nitrogen supplementation on the reproductive success of perennial herbaceous plants, as well as the fundamental mechanisms driving biodiversity in the face of worldwide environmental shifts.

**Abstract:**

The variability observed in the annual seed production of perennial plants can be seen as an indication of changes in the allocation of resources between growth and reproduction, which can be attributed to fluctuations in the environment. However, a significant knowledge gap exists concerning the impacts of nitrogen addition on the interannual seed production patterns of perennial plants. We hypothesized that the addition of nitrogen would impact the annual variations in the seed production of perennial plants, ultimately affecting their overall reproductive efficiency. A multiyear field experiment was conducted to investigate the effects of varying nitrogen supply levels (e.g., 0, 4, and 8 kg N ha^−1^ yr^−1^ of N0, N4, and N8) on vegetative and floral traits, pollinator visitation rates, and seed traits over a period of four consecutive years. The results showed that the N0 treatment exhibited the highest levels of seed production and reproductive efficiency within the initial two years. In contrast, the N4 treatment displayed its highest level of performance in these metrics in the second and third years, whereas the N8 treatment showcased its most favorable outcomes in the third and fourth years. Similar patterns were found in the number of flowers per capitulum and the number of capitula per plant. There exists a positive correlation between aboveground biomass and several factors, including the number of flowers per capitulum, the number of capitula per plant, the volume of nectar per capitulum, and the seed production per plant. A positive correlation was found between pollinator visitation and the number of flowers per capitulum or the number of capitula per plant. This implies that the addition of N affected the maintenance of plant aboveground biomass, flower trait stability, pollinator visitation, and, subsequently, the frequency of seed production and reproductive efficiency. Our results suggest that augmenting the nitrogen content in the soil may have the capacity to modify the inherent variability in seed production that is observed across various years and enhance the effectiveness of reproductive processes. These findings have the potential to enhance our comprehension of the impact of nitrogen addition on the reproductive performance of perennial herbaceous plants and the underlying mechanisms of biodiversity in the context of global environmental changes.

## 1. Introduction

The fundamental principle underlying life-history theory revolves around the notion of trade-offs and the allocation of scarce resources to either reproduction or growth and maintenance necessary for survival [1]. Furthermore, a considerable portion of perennial species demonstrates the ability to undergo multiple cycles of growth and reproduction, enabling them to reproduce across multiple seasons. However, it is imperative to recognize that there is notable year-to-year variability in the seed production of perennial herbs in response to the prevailing growth conditions [1,2]. Plants exhibit enhanced seed production when subjected to favorable years, which encompass optimal soil moisture and nutrient levels as well as suitable weather conditions [3,4]. In light of unfavorable years, there has been a decrease in the overall yield of seeds [1,3]. Thus, there is a diverse range of interannual variations in seed production that is associated with different growth years.

The global distribution of resources is undergoing substantial changes due to increasing levels of carbon dioxide (CO_2_), the deposition of nutrients, and alterations in land use. Predictive models indicate that multiple factors associated with global change have the potential to modify the interannual variation in seed output [3]. The introduction of nitrogen into the soil has a significant influence on the nutrient cycle, particularly in relation to the plant’s ability to obtain readily available resources [5,6]. In theory, the introduction of nitrogen presents a challenge in allocating resources between growth and reproduction, thereby influencing the annual fluctuations in the seed production capacity of perennial plants. As a result, this phenomenon leads to an increase in their overall reproductive efficiency. Nonetheless, there is a notable deficiency in our understanding of the effects of nitrogen supply on the interannual variations in seed production patterns of perennial plants.

The influence of nitrogen on the makeup of soil nutrients controls how it affects the annual fluctuations in reproductive allocation [1,3]. During the early stages of development, plants grown in adequate-N habitats allocate a higher proportion of their resources towards the growth and development of their vegetative organs [7,8]. Subsequently, plants demonstrate an increased allocation of biomass towards their reproductive structures, resulting in an amplified production of flowers [9,10]. Additionally, the influence of nitrogen is a significant factor in the annual fluctuations of plant biomass, as it affects the storage of carbohydrates and the distribution of carbon resources [11,12]. This phenomenon possesses the capacity to induce annual variations in seed production. Additionally, the introduction of nitrogen has the potential to impact both flower production and nectar secretion [13,14,15]. The alterations in floral traits have the potential to influence pollinator foraging behavior, resulting in changes to the reproductive output of self-incompatibility species [9,16]. Therefore, we speculated that the addition of nitrogen would have an impact on the floral traits of perennial plants, leading to fluctuations in seed production and reproductive efficiency on an annual basis.

Based on the available empirical data, it can be deduced that there has been a significant increase in global atmospheric nitrogen inputs [16]. In Europe and North America, the nitrogen (N) addition rate commonly observed ranges from 10 to 25 kg N ha^−1^ year^−1^ [17]. However, it is important to acknowledge that in China, the yearly rate of nitrogen addition is approximately 50 kg N ha^−1^ year^−1^ [18]. The critical load of nitrogen (N) required to elicit a response from alpine meadow communities has been determined to be 10 kg N ha^−1^ yr^−1^ [19]. The wet deposition of nitrogen has been observed to vary across different regions, with 6.69 kg N ha^−1^ yr^−1^ on the western Tibetan Plateau and 7.55 kg N ha^−1^ yr^−1^ on the eastern Tibetan Plateau [20]. The examination of the responses exhibited by high-altitude ecosystems to elevated nitrogen deposition is highly appropriate for investigation on the Tibetan Plateau.

*Saussurea nigrescens* demonstrates self-incompatibility and serves as a dominant herbaceous species in alpine meadows. Our previous study showed that the abundance of honey bees evolved to reduce nectar production [21]. Recently, we observed that the production of its seeds is adversely affected by various environmental factors. Herein, we propose the hypothesis that the addition of nitrogen would have an impact on the annual fluctuations in the seed production of perennial plants, consequently leading to an improvement in their overall reproductive efficacy. In order to evaluate the veracity of this hypothesis, a multiyear field experiment was conducted utilizing *S. nigrescens* in a Tibetan meadow. The species under investigation exhibits a perennial life cycle and is dependent on honey bees for pollination [21]. The objective of this study is to investigate the impacts of nitrogen supplementation on various aspects of plant traits, including the aboveground biomass, patterns of resource allocation, production of flowers and nectar, pollinator visitation, and seed production over multiple years. The expectation was that these factors would collectively have an impact on the interannual variation in seed production. The results of our study hold promise for advancing our understanding of the effects of nitrogen addition on the reproductive efficiency of perennial herbaceous plants, as well as underlying mechanisms that regulate biodiversity in the context of global environmental changes.

## 2. Materials and Methods

### 2.1. Study Site and Species

The research was carried out within the permanent plots of the Hongyuan Alpine Meadow Ecosystem Research Station, which is affiliated with the Chinese Academy of Sciences. The station is situated on the eastern Tibetan Plateau, specifically at coordinates 32°48′–32°52′ N and 102°01′–102°33′ E, with an altitude of 3500 m. The average annual temperature is recorded at 0.9 °C, with the highest temperature occurring in July at 10.9 °C and the lowest temperature occurring in January at −10.3 °C. The yearly average precipitation amounts to 690 mm, with approximately 80% occurring from May to October. In pastures, it is common for sedges, grasses, and forb species to exhibit predominance. The overall vegetation coverage exceeds 95%, while the height of plants remains below 30 cm [21].

The grassland being studied has been subjected to yak grazing for a period of less than twenty years, specifically during the winter seasons. In addition to the practice of cattle grazing, this land does not exhibit any other agricultural uses. Since 1981, beekeepers have been transferring substantial quantities of *Apis mellifera* colonies, which consist of more than 80 million honey bees, to a specifically designated study site area for honey production. The aforementioned colonies are consistently sustained within this region annually, specifically from May to September.

The species under investigation in our study is *S. nigrescens*, which belongs to the perennial herbs of the Asteraceae family. The species frequently exhibit growth patterns within altitudinal ranges spanning from 2900 to 4300 m. The species typically attain reproductive maturity approximately three years after seed germination. The height of the plants ranges from 15 to 40 cm, and they produce 2 to 5 capitula. Each capitulum contains 20 to 55 florets, specifically mono-ovulated florets. The annular nectary is composed of a bowl-shaped tissue located at the upper region of the ovary, positioned between the ovary and the anther. The bowl’s diameter ranges from 1.5 to 2.0 mm, while the style’s diameter ranges from 0.3 to 0.5 mm. The observed species exhibits self-incompatibility and initiates its growth phase in the middle of May annually while undergoing senescence in the middle of September [21]. Plants frequently undergo the process of flowering during the period spanning from late July to August, followed by the subsequent development of fruits in early September. Apiculture has been consistently practiced in the alpine meadows of the Zoige Plateau, located in the eastern region of the Tibetan Plateau, since 1981. This technique involves the utilization of the honey bee species known as *Apis mellifera*. From late July to early August, honey bees engaged in foraging activities exhibited a substantial reliance on *S. nigrescens* as a primary source of nectar collection. It is worth noting that the reproductive success of *S. nigrescens* is significantly influenced by the presence of *A. mellifera* [21,22].

### 2.2. Nitrogen Addition Experiment

In this study, we present three levels of nitrogen (0, 4, and 8 g N m^−2^ yr^−1^) addition experiments conducted in an alpine meadow located on the eastern Tibetan Plateau [18]. The study employed a fully randomized block design, incorporating three distinct treatments: control (0 g N m^−2^ yr^−1^), N4 (4 g N m^−2^ yr^−1^), and N8 (8 g N m^−2^ yr^−1^). The three treatments were allocated randomly within a block design and replicated six times [23]. A matrix configuration comprising 18 plots (2 m × 2 m) was established. The arrangement of the plots adhered to a 3 × 6 grid pattern. The distance between adjacent plots was measured to be 3 m. The nitrogen supply was carried out using urea. In light of the frequent incidence of N wet deposition, particularly during the summer season in the specified geographical area, a solution comprising urea was administered to the plots via a sprayer, specifically in the initial week of May [18]. Following the fertilization process, the vegetation underwent thorough irrigation using water. The cumulative amount of water dispensed corresponded to approximately 2 m of precipitation. In the experimental configuration, a standardized quantity of water was evenly dispersed among the control plots [18]. Fertilization was applied to the experimental plots during the corresponding growing seasons from 2019 to 2022.

### 2.3. Measurements of Vegetative and Flower Trait

In late July 2018, 8–10 seedlings were randomly selected for each plot. Following that, a protective framework comprising six iron wires measuring 1 mm in diameter was constructed around each plant, which stood at a height of 5 m. During the initial flowering period, 5 plants of selection in each plot were labeled with tags. During the period of peak flowering, we measured the number of capitula per plant as well as the number of flowers per capitulum from 2019 to 2022 [21].

In order to quantify the volume and concentration of nectar, a method was employed where selected plants were subjected to a 24-h period of coverage using cylindrical metal netting [24]. This was performed with the intention of preventing pollinators’ access to the plants under investigation. The volume of nectar per flower was measured using micropipettes with a capacity of 1 or 2 µL. Concurrently, a hand refractometer with a precision of 0.5% was employed to quantify the concentration of nectar (Eclipse, Stanley, and Bellingham, Basingstoke, UK). Measurements were conducted from 08:00 to 14:30 on sunny days. Based on our empirical observations, it was found that the presence of pale to white anthers in flowers was indicative of the highest level of nectar volume production [21]. We conducted measurements on a range of 3–5 flowers per capitulum. Nectar volume per capitulum was calculated as nectar volume per flower multiplied by the number of capitula per plant. The number of flowers per plant was calculated as the number of capitula per plant with multiple numbers of flowers per capitulum.

At the end of the experiment, the aerial parts of each plant were collected and divided into leaf, capitulum, and stem. The plant materials that were collected were segregated into individual paper bags, over-dried at 75 °C for 48 h, and subsequently measured using a precision balance with an accuracy of 0.001 g.

### 2.4. Measurements of Pollinator Visitation

Field observations were carried out during the peak blooming period of *S. nigrescens*, spanning from 2019 to 2022, with the aim of investigating the rates of pollinator visitation. The capitula number of each sampled plant in each plot was initially recorded. Honey bee visitation to the capitula was monitored by observers at a distance of 3 m. The observation periods were evenly distributed throughout the day, occurring between 9:00 and 17:00. During each hour, a single observer viewed each plant for a duration of two minutes from a stationary position. There were three observers, resulting in each plant being observed twice per hour. The observations documented the number of capitula for each plant that was visited by honey bees during the observation period [25]. The observations were exclusively conducted during sunny days and were repeated nine days per plant per year, spanning from 2019 to 2022. A cumulative duration of 144 min of field observations was recorded annually for each plant. Visitation rates (R) per capitulum per hour were calculated as the total number of visits (Nv) per hour divided by the number of capitula (Nc), i.e., R = Nv/Nc [26].

### 2.5. Measurements of Seed Traits

In the middle of September, ripened capitula were dissected to determine the number of ovules, unfertilized ovules, and viable seeds (which corresponded to the number of ovules per capitulum). The ratio of viable seeds to the total number of ovules was referred to as the seed set, i.e., seed set = seed number per capitulum/total ovule number per capitulum [21]. The number of viable seeds stands for the number of seeds per capitulum. The seeds per capitulum were also measured using a balance (0.001 g) as seed yield.

Seed yield per plant was calculated as seed yield per capitulum multiplied by the number of capitula per plant. Reproductive efficiency was calculated as the seed yield per plant divided by the sum of the aboveground biomass.

### 2.6. Data Analysis

For analysis, the data pertaining to each trait were first averaged for every plot in this study. The statistical analyses were conducted using the R software 4.2.1 (R Development Core Team 2022). The R software can be accessed at http://www.R-project.org/ (accessed on 23 June 2022).

The data were analyzed using generalized linear mixed models (GLMMs). The fixed effects in the study encompassed N addition treatments (N0, N4, and N8) and the experimental years spanning from 2019 to 2022, while the plot was considered a random effect. The above-ground biomass, nectar concentration, and pollinator visitation rate were modeled using a Gaussian model with an identity-link function. Next, we employed a Gamma model (with a Logit-link) and utilized the Laplace approximation to estimate parameters in order to model various factors, including the number of capitula per plant, number of flowers per capitulum or per plant, nectar volume per capitulum or per plant, seed set, number of seeds per capitulum or per plant, reproductive efficiency, and reproduction allocation. The parameters were estimated using the Laplace approximation method, specifically utilizing the glm.nb function in the ‘lme4’ package. Upon examination of the residuals, it was determined that a normal error distribution was suitable. Following the identification of a notable impact on the N addition and the experimental year, subsequent post-hoc LSD tests were employed to conduct pairwise comparisons between the N addition and the years. Calculations were performed using the glmer function from the lme4 package [27].

The study aimed to examine the association between flower traits and pollinator visitation rate at the individual plant level. This investigation utilized the automated model selection method provided by the R package glmulti. Pollinator visitation rates can be influenced by several factors, including the nectar volume per capitula, nectar concentration, number of flowers per plant, and above-ground biomass. In this study, we used a random effect model with a random intercept for each plot (random = 1|plot) to account for potential variability between plots. The authors of the study conducted an analysis to determine the relative significance of each predictor in influencing the outcome [28]. A threshold value of 0.8 is needed in order to identify the variables deemed most significant [29].

The R package “lares” was utilized to examine the ranked cross-correlations among all plant and pollinator variables (http://laresbernardo.github.io/lares/reference/corr_cross.html, accessed on 23 June 2022).

## 3. Results

Interannual vegetative and floral traits of *S. nigrescens* were found to be regulated by soil nitrogen levels (Figure 1 and Figure 2, and Table 1 and Table 2). The N4 and N8 treatments exhibited the greatest aboveground biomass, stem mass, and capitulum mass in the third and fourth years. In contrast, no significant variation in aboveground biomass was observed across the years for the N0 treatment (Figure 1). The number of capitula per plant and the number of flowers per capitulum reached their highest levels during the second and third years for the N4 treatment and in the third and fourth years for the N8 treatment. The highest recorded nectar volume per capitulum was observed in the first three years for the N0 treatment, while for the N8 treatment, it was observed in the third and fourth years. On the other hand, there was no statistically significant variation in nectar volume per capitulum for the N4 treatment across different years (Figure 2). The addition of nitrogen did not result in any year-to-year variation in nectar concentration (Figure 2).

The introduction of nitrogen had an impact on both the rate of pollinator visitation and seed set (Figure 3, Table 3). Peak pollinator visitation and seed set were observed during the initial two years for the N0 treatment and in the third and fourth years for the N8 treatment. Conversely, no discernible differences in these variables were found during the years of the N4 treatment (Figure 3).

The nitrogen supply in the soil was seen to have an impact on both reproductive efficiency and seed production per capitulum or plant (Figure 4a,b and Figure 5; Table 3). The N0 treatment exhibited the highest reproductive efficiency and the greatest number of seeds per capitulum during the initial two years. Similarly, the N4 treatment demonstrated these characteristics in the second and third years, while the N8 treatment displayed them in the third and fourth years.

The introduction of nitrogen had varying effects on the correlations between different traits (Figure 6). The study revealed a positive correlation between seed production per plant and both flower production per capitulum and capitulum production per plant (Figure 6). There exists a positive correlation between the number of seeds per capitulum, the frequency of pollinator visits, and the quantity of nectar produced by each capitulum. The study revealed a positive correlation between above-ground biomass and various factors, including reproductive efficiency, seed set, pollinator visitation frequency, and the number of capitula per plant (Figure 6).

Both the overall quantity of capitula per plant and the number of seeds per capitulum exhibited a noteworthy influence on seed production. (Figure 4c). Both the level of nitrogen addition and the experimental year had an impact on the number of seeds that a plant produced (Figure 4c).

The results of model selection indicated a positive relationship between pollinator visitation rates and the number of flowers and capitula on a plant (Figure 7).

## 4. Discussion

The findings indicated that increasing the nitrogen content in the soil has the potential to alter the natural fluctuations in seed production and reproductive efficiency that occur over different years of *S. nigrescens*. Nitrogen has been found to alter the interannual variations in the number of flowers per capitulum as well as the number of capitula per plant, consequently affecting the annual frequency of seed production. However, the introduction of nitrogen had an impact on the fluctuation of flower rewards from year to year. This included changes in nectar production per capitulum and the number of flowers per plant. Additionally, the patterns of pollinator visitation rate and seed set were also altered throughout the duration of the experiment. A positive correlation has been observed between the number of seeds produced per plant and several factors, such as the number of flowers per capitulum, the number of capitula per plant, the volume of nectar per capitulum, and the rate of visitation by pollinators. A significant positive correlation was identified between aboveground biomass and several factors, encompassing the allocation of biomass to the stem, the number of flowers per capitulum, the number of capitula per plant, the volume of nectar per capitulum, and the seed production per plant. This demonstrates that the incorporation of element N significantly influences the maintenance of aboveground biomass and flower traits, resulting in alterations in the frequency of pollinator visits and the interannual variability in seed production. The findings of this research have the potential to enhance our comprehension of the impacts of nitrogen supply on the reproductive efficacy of perennial herbs as well as the underlying mechanisms that influence biodiversity in alpine meadows amidst global environmental changes.

The presence of nitrogen significantly influences plant biomass through the alteration of carbohydrate reserves and the allocation of carbon resources [11,12]. Plants grown in habitats with an adequate nitrogen supply often exhibit a greater allocation of biomass towards the stem [30,31]. This allocation is necessary to provide mechanical support for the increased aboveground biomass of *S. nigrescens*. In addition, the introduction of nitrogen resulted in an increase in community height (30~35 cm). This increase in height can be attributed to the promotion of stem growth, which enables *S. nigrescens* to compete better for light resources [32,33]. Moreover, the highest aboveground biomass was observed during the third and fourth years of our study (Figure 1). We postulated that a higher percentage of underground resources experienced a transformation resulting in the production of a greater amount of aboveground biomass over the course of the latter two years [31].

Furthermore, plants frequently allocate a greater number of resources towards the development of their reproductive organs, including the production of a larger quantity of flowers and the provision of increased flower rewards [9,10,34]. This relationship is exemplified by a positive association between the aboveground biomass and several floral traits, such as the number of capitula per plant, the number of flowers per capitulum, and the amount of nectar per capitulum (Figure 6). Both the N4 and N8 treatments exhibited an increase in aboveground biomass as well as an increase in the flower mass fraction (Figure 1). This implies that an augmentation in aboveground biomass has the potential to enhance the number of flowers and nectar [13,35]. Furthermore, our study revealed a positive correlation between nectar and flower production and aboveground biomass fluctuations over consecutive years. The flower or capitulum number and nectar production of the plant and capitulum exhibit interannual variation, which is influenced by changes in aboveground biomass over the course of the experimental years. This finding suggests that the addition of nitrogen not only alters the year-to-year fluctuations in aboveground biomass [31] but also affects the allocation of resources towards flower and nectar production, consequently leading to annual changes in flower and nectar abundance.

In a research investigation examining the impacts of nutrient supply on nectar traits, the addition of nitrogen did not result in an increase in nectar secretion for *Trifolium pratense*. However, it did lead to an increase in the rate of nectar secretion for *Antirrhinum majus* [36]. Nevertheless, the augmentation in nectar secretion of *A. majus* and *Ipomopsis aggregata* was solely observed under conditions of minimal nitrogen supplementation (10 kg N ha^−1^ year^−1^). At elevated levels of nitrogen addition (200 kg N ha^−1^ year^−1^), there was a notable reduction in nectar secretion for both species [13,36]. This suggests that the influence of nitrogen supplementation on nectar production is dependent on both the dosage of nitrogen and the particular plant species. The nectar secretion of *S. nigrescens* was found to be enhanced by both the N4 and N8 treatments. The extent to which the dose of nitrogen reduces nectar production in *S. nigrescens* remains unknown. Indeed, numerous studies have demonstrated that the introduction of nitrogen has a positive impact on the overall concentration of amino acids [37,38,39]. The presence of nitrogen addition has been frequently observed to result in an increase in the levels of asparagine and glutamine among the various individual amino acids [15,37,39]. The disparity in nectar concentration (e.g., sugar content) was not identified in *S. nigrescens*. The concentration of nectar is subject to influence by microclimate factors, particularly relative humidity [40]. The available evidence indicates that the amount of water in nectar often affects how concentrated it is. The humidity gradient affects the rate of evaporation, which in turn affects the flow of water between nectar and air [41], which in turn affects the amount of water in nectar. The study site where three treatments were conducted exhibited similar levels of humidity. The plants exhibited uniform growth under equivalent relative humidity conditions, leading to nectar concentrations that were indistinguishable.

Bees derive their energy from the consumption of nectar and pollen. The provision of protein and other essential nutrients is imperative for the optimal growth and development of larvae. Plants that exhibit a high production of nectar in substantial quantities have the potential to attract a greater number of visits from pollinators as well as prolong the duration of the visits [21,42,43]. Within the designated study areas, it has been observed that honey bees play a predominant role as the principal pollinators of *S. nigrescens*. From late July to early August, domesticated honey bees demonstrate a notable reliance on *S. nigrescens* as their principal nectar-gathering resource [21]. There was a correlation between honey bee visitation and nectar production [44]. Plants that thrive in N4 and N8 habitats, characterized by the presence of nectar-rich flowers, exhibit a higher frequency of pollinator visits. Conversely, plants inhabiting the N0 habitat, which is characterized by the absence of nectar-rich flowers, tend to attract fewer pollinators (Figure 2 and Figure 3). A positive correlation has been observed between the overall sugar concentration in nectars and specific pollinators [45]. As an illustration, honey bees exhibit a preference for nectar characterized by a diminished sugar concentration [46], while bumble bees display a preference for nectar characterized by an elevated sugar concentration [47]. The sugar concentration of *S. nigrescens* exhibited a range of 37% to 50%, aligning with the nectar concentration preferences observed in honey bees. The level of flower production has an impact on the frequency of flower visits. The findings from our model selection analysis indicated that both the quantity of nectar produced per plant and the number of capitula per plant are significant factors in determining the visitation of pollinators. It is imperative to acknowledge that floral scents have a significant impact on the attraction of pollinators [48]. Nevertheless, the impact of nitrogen supplementation on the olfactory traits of flowers and the preferences exhibited by pollinators remains uncertain. The visitation rate is contingent upon the combined factors of species richness and the abundance of pollinators [49]. The honey bee population size exhibited no discernible variation among the study sites as a result of the substantial release of honey bees by beekeepers in the designated study areas [21,22].

The introduction of nitrogen has the potential to modify the inherent variations in seed production and reproductive efficiency observed across different years in *S. nigrescens*. The presence of nitrogen has been observed to have an impact on the interannual variations in both the number of flowers per capitulum and the number of capitula per plant (Figure 6). The seed yield per plant is influenced by two factors: the number of capitula per plant and the number of seeds per capitulum (Figure 4). *S. nigrescens* exhibits self-incompatibility, with honey bees serving as the primary pollinators for fertilization [21]. This deduction is supported by the demonstrated positive correlation between seed set and pollinator visitation. The introduction of nitrogen resulted in a notable augmentation in the number of flowers and capitula, as well as an increase in nectar production and pollinator visitation [13,14,16], consequently leading to an enhanced seed output. Moreover, there is a correlation between aboveground biomass and the annual variation in seed production. A positive correlation has been identified between above-ground biomass and various reproductive traits, including seed quantity per plant, flower quantity per capitulum, and capitulum quantity per plant. This implies that the addition of nitrogen has the potential to modify the year-to-year fluctuations in aboveground biomass, resulting in changes in the number of flowers per capitulum, the number of capitula per plant, nectar production, and pollinator visitation across various years. Consequently, this can also impact interannual seed output and reproductive efficiency.

Interannual variations in seed yield exert a significant impact on the sustainability of perennial plant populations [1,31]. In light of an uncertain environment, plants undertake a thorough evaluation of the benefits and costs linked to various reproductive strategies. Under optimal circumstances, plants exhibit a tendency to produce a larger number of seeds, thereby providing benefits for the replenishment of populations. Plants demonstrate a reduction in resource allocation towards reproduction when confronted with adverse circumstances [50], opting to prioritize resource allocation towards survival. Our study suggests that over the course of the first two years of nitrogen supplementation, plants exhibited a preference for allocating resources toward vegetative tissue [51]. This allocation strategy ultimately resulted in a significant improvement in seed yields during the subsequent third and fourth years for N4 and N8 plants.

The presence of soil nutrients has the potential to influence reproductive processes [9]. For example, the presence of mycorrhizal infection in soil with high phosphorus content has the potential to enhance seed production [52]. The present study did not observe the variation in soil nitrogen (N) and the other element levels among the different nitrogen supply treatments. It was found that soil nutrients exerted a notable impact on the secretion and concentration of nectar in flowers. Consequently, these factors influenced the rates of visitation by pollinators and the production of seeds [13]. Further investigation would be required to conduct a comprehensive examination. It has been seen that adding nitrogen changes the composition and activity of nitrogen-fixing bacteria [53]. This has an effect on how seed production changes from year to year. Further investigation is warranted to empirically examine the potential impacts of this phenomenon.

In general, increasing the amount of nitrogen in the soil can change the natural differences in seed yield and improve the reproductive success of *S. nigrescens* seen in different years. The augmentation of nitrogen concentrations has a positive impact on the seed production of *S. nigrescens*, especially when combined with a dosage of 8 g·m^−2^ of nitrogen. However, the potential effects of nitrogen addition on the processes of seed germination and seeding establishment have yet to be investigated. The impact on plant abundance and community composition remains uncertain. The results of this study could help us learn more about how nitrogen supplementation affects the ability of perennial herbaceous plants to reproduce, as well as the basic processes that affect biodiversity in alpine meadows when the global environment changes.

## Figures and Tables

**Figure 1 biology-12-01132-f001:**
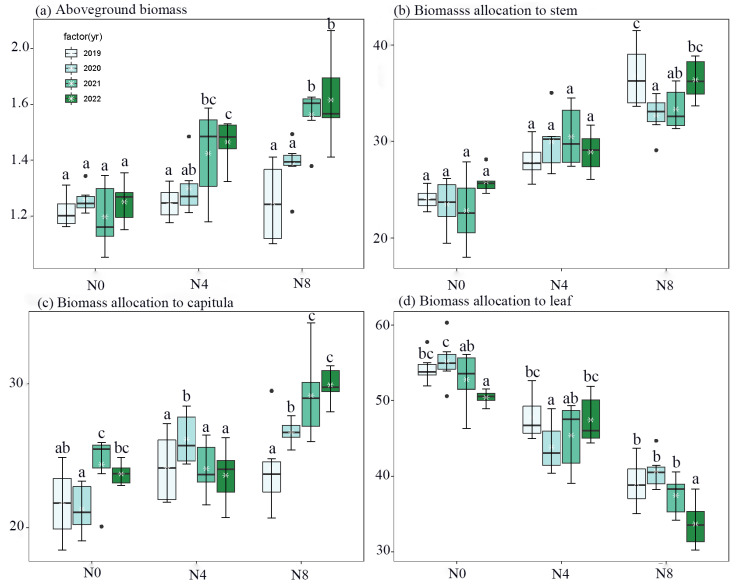
Comparison of aboveground biomass (**a**), biomass allocation to stem (**b**), capitulum (**c**), and leaf (**d**) across N0, N4, and N8 treatments for each year (from 2019 to 2022) of *Saussurea nigrescens*. Different letters above the boxes denote significant differences among treatments (*p* < 0.05).

**Figure 2 biology-12-01132-f002:**
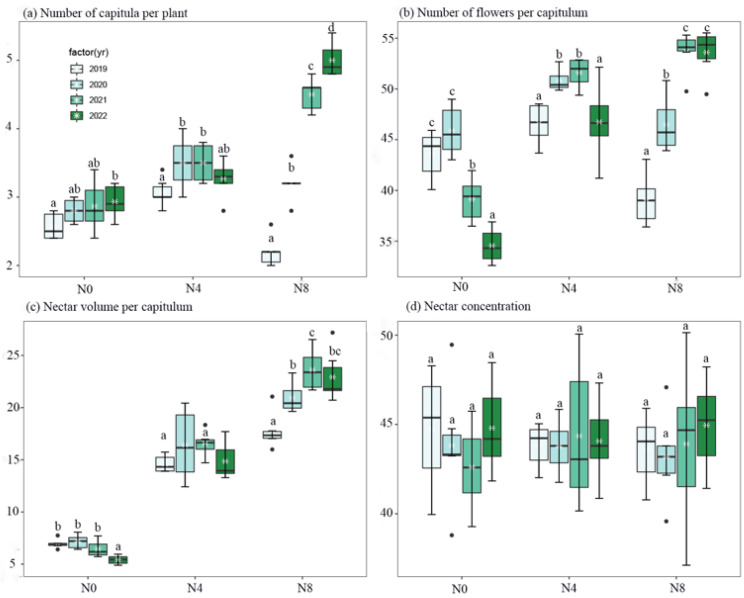
Comparison of the number of capitula per plant (**a**), number of flowers per capitula (**b**), nectar volume per capitula (**c**), and nectar concentration (**d**) across N0, N4, and N8 treatments for each year (from 2019 to 2022) of *Saussurea nigrescens*. Different letters above the boxes denote significant differences among treatments (*p* < 0.05).

**Figure 3 biology-12-01132-f003:**
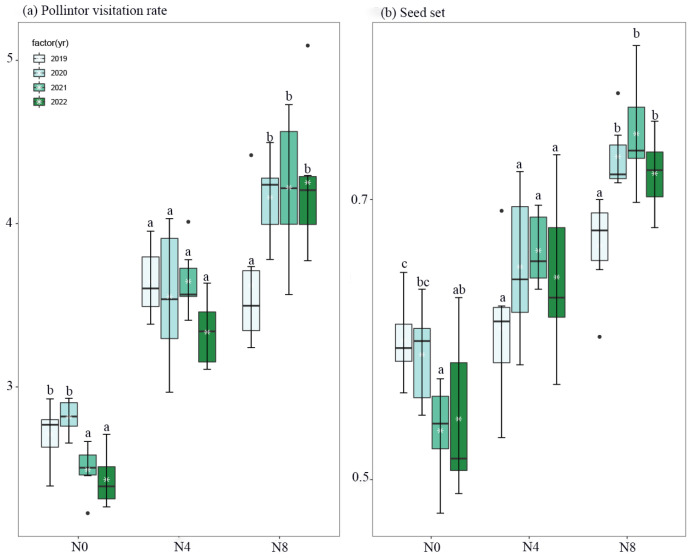
Comparison of the number of pollinator visitation rate (**a**) and seed set (**b**) across N0, N4, and N8 treatments for each year (from 2019 to 2022) of *Saussurea nigrescens*. Different letters above the boxes denote significant differences among treatments (*p* < 0.05). ssr—seed set; vr—pollination visitation rate (visitors capitula^−1^·hr^−1^).

**Figure 4 biology-12-01132-f004:**
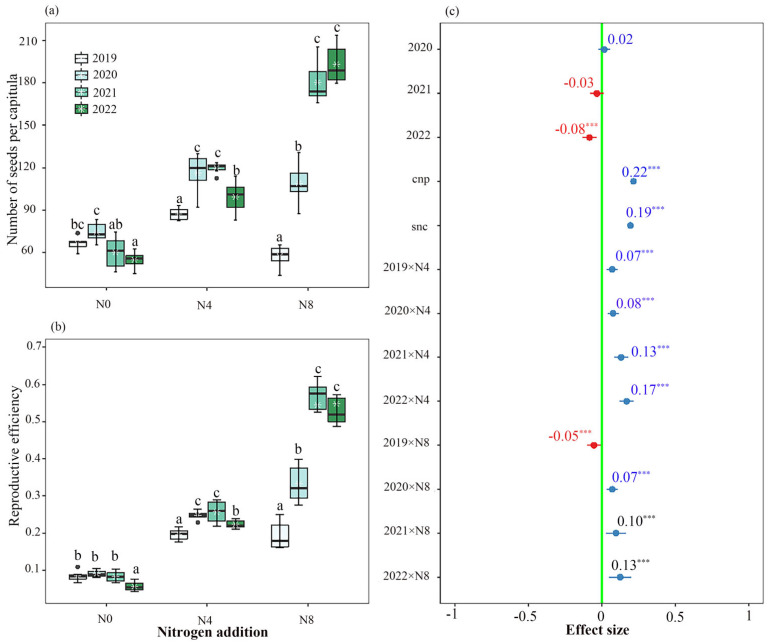
Interannual variation in the number of seeds produced per plant (**a**) and the reproductive efficiency (**b**) of *Saussurea nigrescens* among the N0 (0 g·m^−2^ N), N4 (4 g·m^−2^) and N8 (8 g·m^−2^) treatments; additionally, the effects of nitrogen addition and each experimental year on the number of seeds per plant (**c**) was analyzed using generalized linear mixed models (GLMMs). Different letters above the boxes denote significant differences among treatments (*p* < 0.05). The effect of nitrogen addition and each experimental year on the number of seeds per plant indicated by values and 95% confidence interval (CI) of estimated slopes from GLMM, with significant (95% CI does not overlap with zero, *** *p* < 0.001) positive (blue) and negative (red) effect highlighted by colored values with * and CI of the estimated slopes, and values without * and CI (overlap with zero) indicated non-significant difference across the N0, N4, and N8 treatments. The variable “cnp” represents the number of capitula per plant, while the variable “snc” represents the number of seeds per capitula.

**Figure 5 biology-12-01132-f005:**
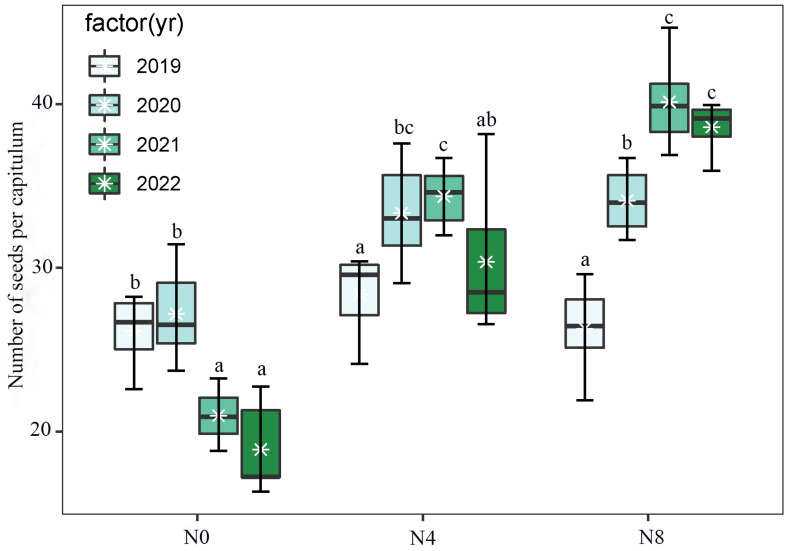
Comparison of the number of seeds per capitula across N0, N4, and N8 treatments for each year (from 2019 to 2022) of *Saussurea nigrescens*. Different letters above the boxes denote significant differences among treatments (*p* < 0.05).

**Figure 6 biology-12-01132-f006:**
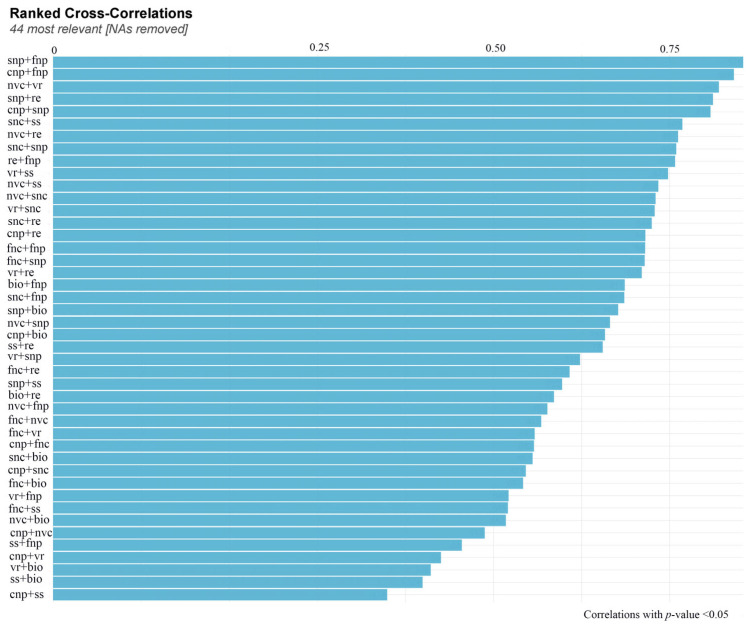
Ranked cross-correlations for variables of vegetative and floret traits, seed traits, and pollinator visitation rate of *Saussurea nigrescens*. Blue colors correspond to positive correlations. The size of the bar was proportional to the correlation coefficients. Forty-four most relevant reflected significant correlations of variables. NAs removed reflect the insignificant correlations that did not present. bio—above-ground biomass; cnp—number of capitula per plant; fnc—number of flowers per capitula; fnp—number of flowers per plant; nc—nectar concentration; nvc—nectar volume per capitula; nvp—nectar volume per plant; snc—number of seeds per capitula; snp—number of seeds per plant; ssr—seed set; vr—pollination visitation rate; nvc + vr represents the relationship between nectar volume per capitula and pollinator visitation rate.

**Figure 7 biology-12-01132-f007:**
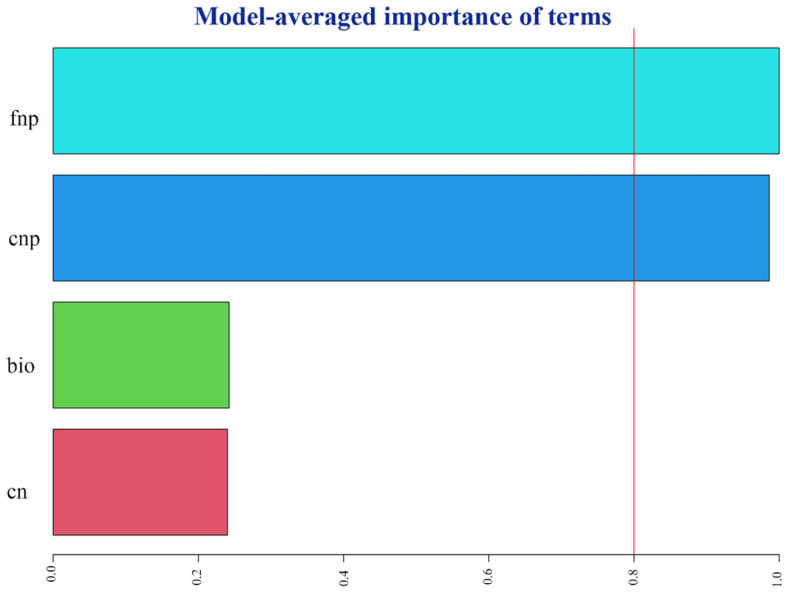
Relative importance of flower traits on pollinator visitation rate of *Saussurea nigrescens*. The red line on the chart indicates which terms are significant. fnp—number of flowers per plant; cnp—number of capitula per plant; bio—above-ground biomass; nc—nectar concentration. The figure shows the result of the best model selection.

**Table 1 biology-12-01132-t001:** Effects of nitrogen addition and each year on vegetative and flower traits.

	Aboveground Biomass			Number of Capitulum per Plant			Number of Flowers per Capitula			Number of Flowers per Plant		
Predictors	Estimates	CI	*p*	Estimates	CI	*p*	Estimates	CI	*p*	Estimates	CI	*p*
Intercept	1.42	1.33–1.50	<0.001	2.57	2.40–2.74	<0.001	43.56	41.87–45.33	<0.001	111.68	103.47–120.54	<0.001
2020	0.04	−0.06–0.15	0.432	1.09	1.00–1.18	0.039	1.05	1.00–1.11	0.050	1.14	1.04–1.25	0.005
2021	−0.02	−0.13–0.09	0.719	1.12	1.03–1.21	0.010	0.90	0.85–0.94	<0.001	1.00	0.92–1.10	0.934
2022	0.03	−0.07–0.14	0.515	1.14	1.05–1.24	0.002	0.79	0.75–0.83	<0.001	0.91	0.83–0.99	0.034
tr [N4]	0.03	−0.08–0.14	0.568	1.20	1.10–1.30	<0.001	1.07	1.02–1.13	0.011	1.28	1.17–1.40	<0.001
tr [N8]	0.03	−0.07–0.14	0.556	0.86	0.79–0.93	<0.001	0.90	0.85–0.94	<0.001	0.77	0.70–0.84	<0.001
2020 × tr [N4]	0.01	−0.14–0.16	0.882	1.05	0.93–1.17	0.445	1.04	0.96–1.11	0.328	1.09	0.96–1.24	0.172
2021 × tr [N4]	0.20	0.05–0.35	0.011	1.02	0.91–1.15	0.704	1.23	1.15–1.33	<0.001	1.26	1.11–1.43	0.001
2022 × tr [N4]	0.18	0.03–0.33	0.018	0.93	0.83–1.05	0.227	1.26	1.18–1.36	<0.001	1.18	1.04–1.34	0.013
2020 × tr [N8]	0.09	−0.06–0.24	0.214	1.33	1.19–1.50	<0.001	1.13	1.05–1.21	0.001	1.52	1.34–1.72	<0.001
2021 × tr [N8]	0.33	0.18–0.48	<0.001	1.83	1.63–2.06	<0.001	1.53	1.42–1.64	<0.001	2.80	2.47–3.18	<0.001
2022 × tr [N8]	0.33	0.18–0.48	<0.001	1.99	1.77–2.23	<0.001	1.73	1.61–1.86	<0.001	3.44	3.03–3.90	<0.001

**Table 2 biology-12-01132-t002:** Effects of nitrogen addition and each year on nectar volume and concentration.

	Nectar Concentration			Nectar Volume per Capitula			Nectar Volume per Plant		
Predictors	Estimates	CI	*p*	Estimates	CI	*p*	Estimates	CI	*p*
Intercept	44.72	42.60–46.84	<0.001	6.95	6.40–7.55	<0.001	18.10	16.18–20.25	<0.001
2020	−0.90	−3.90–2.09	0.548	1.03	0.93–1.14	0.585	1.09	0.96–1.24	0.184
2021	−2.12	−5.11–0.88	0.162	0.93	0.84–1.03	0.150	1.03	0.90–1.17	0.667
2022	0.08	−2.91–3.07	0.958	0.78	0.70–0.86	<0.001	0.87	0.76–0.99	0.030
tr [N4]	−0.89	−3.88–2.10	0.554	2.10	1.90–2.32	<0.001	2.47	2.17–2.81	<0.001
tr [N8]	−1.11	−4.10–1.88	0.461	2.56	2.31–2.83	<0.001	2.14	1.88–2.43	<0.001
2020 × tr [N4]	0.84	−3.39–5.08	0.691	1.09	0.94–1.26	0.233	1.17	0.98–1.40	0.090
2021 × tr [N4]	2.64	−1.59–6.87	0.217	1.22	1.05–1.40	0.008	1.25	1.04–1.50	0.017
2022 × tr [N4]	0.16	−4.08–4.39	0.941	1.31	1.13–1.51	<0.001	1.25	1.05–1.50	0.015
2020 × tr [N8]	0.46	−3.77–4.69	0.829	1.15	0.99–1.32	0.061	1.58	1.31–1.89	<0.001
2021 × tr [N8]	2.42	−1.82–6.65	0.258	1.43	1.24–1.65	<0.001	2.68	2.23–3.21	<0.001
2022 × tr [N8]	1.26	−2.97–5.50	0.553	1.66	1.44–1.91	<0.001	3.42	2.85–4.10	<0.001

**Table 3 biology-12-01132-t003:** Effects of nitrogen addition and each year on visitation rate, seed set, number of seeds per capitula, and reproductive efficiency.

	Pollinator Visitation Rate			Seed Set			Number of Seeds per Capitula			Reproductive Efficiency		
Predictors	Estimates	CI	*p*	Estimates	CI	*p*	Estimates	CI	*p*	Estimates	CI	*p*
Intercept	2.71	2.48–2.93	<0.001	0.60	0.57–0.63	<0.001	26.14	24.33–28.09	<0.001	0.08	0.08–0.09	<0.001
2020	0.11	−0.21–0.42	0.495	0.98	0.92–1.06	0.640	1.04	0.94–1.15	0.444	1.07	0.92–1.25	0.364
2021	−0.22	−0.53–0.10	0.173	0.89	0.83–0.96	0.002	0.80	0.72–0.89	<0.001	0.98	0.84–1.14	0.789
2022	−0.28	−0.59–0.04	0.083	0.91	0.84–0.97	0.008	0.72	0.65–0.80	<0.001	0.67	0.58–0.78	<0.001
tr [N4]	0.94	0.62–1.25	<0.001	1.01	0.94–1.09	0.689	1.09	0.98–1.20	0.110	2.32	1.99–2.70	<0.001
tr [N8]	0.91	0.60–1.23	<0.001	1.11	1.04–1.20	0.004	1.01	0.91–1.11	0.921	2.28	1.96–2.65	<0.001
2020 × tr [N4]	−0.20	−0.64–0.25	0.377	1.09	0.99–1.21	0.092	1.13	0.98–1.30	0.096	1.18	0.95–1.46	0.126
2021 × tr [N4]	0.22	−0.22–0.66	0.326	1.22	1.11–1.35	<0.001	1.51	1.31–1.74	<0.001	1.34	1.08–1.65	0.009
2022 × tr [N4]	−0.03	−0.48–0.41	0.885	1.17	1.06–1.29	0.003	1.48	1.28–1.71	<0.001	1.70	1.37–2.11	<0.001
2020 × tr [N8]	0.43	−0.01–0.87	0.056	1.11	1.01–1.23	0.037	1.25	1.08–1.44	0.003	1.61	1.30–1.99	<0.001
2021 × tr [N8]	0.82	0.37–1.26	0.001	1.25	1.13–1.39	<0.001	1.90	1.65–2.20	<0.001	2.88	2.33–3.57	<0.001
2022 × tr [N8]	0.91	0.46–1.35	<0.001	1.19	1.07–1.32	0.001	2.03	1.76–2.34	<0.001	4.24	3.42–5.25	<0.001

## Data Availability

Not applicable.

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
