# Peer review of "Nitrogen Addition Affects Interannual Variation in Seed Production in a Tibetan Perennial Herb"

_biology, 2023, doi:10.3390/biology12081132_

Round 1

Reviewer 1 Report

Reviewer comments

Manuscript: biology-2548212 - Nitrogen addition affects interannual variation in seed production in a Tibetan perennial herb

The authors examined the impact of nitrogen addition on the interannual seed production patterns of perennial plants. The introduction of element N had a significant impact on the preservation of aboveground biomass in plants, as well as the stability of flower traits. The authors indicated that increasing the nitrogen content in the soil has the potential to alter the natural fluctuations in seed production that occur over different years. The results of their study possess the capacity to improve the understanding of the effects of nitrogen supplementation on the reproductive success of perennial herbaceous plants, as well as the fundamental mechanisms driving biodiversity in the face of worldwide environmental shifts.

The statistical analysis and methods are correct.

English language and style are fine and minor spell check required.

The uniqueness of the text is more than 90% by AntiPlagiarism.NET.

There are some comments and questions:

1) It has long been known to everyone that Nitrogen is a growth stimulant of the plants. Nitrogen helps to form a strong plant structure and to increase green mass and yield. Please explain what is novelty and uniqueness of your study.

2) Lines 177-178 - Measurements were conducted during the time interval of 08:00 to 14:30. Please explain how this interval was selected.

3) There is no discussion about nitrogen-fixing bacteria in introduction and discussion parts. Authors should check the presence of Nitrogen-fixing bacteria in the soil to evaluate final effect on the plants.

4) The numbers in chemical formulas should be subscripted. For example, Lines 155 and 157.

5) Line 102 - first mention of S. nigrescens must be written in full - Saussurea nigrescens. 

5) Lines 102-103 - The field experiment was conducted utilizing the S. nigrescens in Tibetan meadow. Explain why you selected this plant model. 

6) Line 118 - with an altitude of 3500 m. Explain why you studied this place.

7) Line 396 - absence of nectar-poor flowers - it should be probably - absence of nectar-rich flowers.

8) Explain the use as pollinator honey bees Apis mellifera.

9) How yak grazing changed the grassland?

10) Lines 373-374 =  The extent to which the dose of nitrogen reduces nectar production remains unknown. - It is already known - Cite this article - Vaudo, A.D., Erickson, E., Patch, H.M. et al. Impacts of soil nutrition on floral traits, pollinator attraction, and fitness in cucumbers (Cucumis sativus L.). Sci Rep 12, 21802 (2022). https://doi.org/10.1038/s41598-022-26164-4

Please improve the manuscript according to the above comments.

English language and style are fine and minor spell check required.

Author Response

Reviewer 1

Manuscript: biology-2548212 - Nitrogen addition affects interannual variation in seed production in a Tibetan perennial herb

The authors examined the impact of nitrogen addition on the interannual seed production patterns of perennial plants. The introduction of element N had a significant impact on the preservation of aboveground biomass in plants, as well as the stability of flower traits. The authors indicated that increasing the nitrogen content in the soil has the potential to alter the natural fluctuations in seed production that occur over different years. The results of their study possess the capacity to improve the understanding of the effects of nitrogen supplementation on the reproductive success of perennial herbaceous plants, as well as the fundamental mechanisms driving biodiversity in the face of worldwide environmental shifts.

The statistical analysis and methods are correct. English language and style are fine and minor spell check required.

There are some comments and questions:

  • It has long been known to everyone that Nitrogen is a growth stimulant of the plants. Nitrogen helps to form a strong plant structure and to increase green mass and yield. Please explain what is novelty and uniqueness of your study.

Response 1. The results of our study suggest that manipulating the nitrogen levels in the soil may have an impact on the inherent variability in seed production and reproductive efficiency observed across different years in S. nigrescens. The variability of flower rewards from year to year, as well as the patterns of pollinator visitation rate and seed set, were influenced by the introduction of nitrogen. The findings of our study indicate that the inclusion of nitrogen (N) element has a substantial impact on the preservation of aboveground biomass and flower traits. This, in turn, leads to changes in the frequency of pollinator visits and the year-to-year fluctuations in seed output. This was stressed in the initial paragraph of the discussion.

  • Lines 177-178 - Measurements were conducted during the time interval of 08:00 to 14:30. Please explain how this interval was selected.

Response 2. We rewrote this sentence. “Measurements were conducted from 08:00 to 14:30 on sunny days”.

  • There is no discussion about nitrogen-fixing bacteria in introduction and discussion parts. Authors should check the presence of Nitrogen-fixing bacteria in the soil to evaluate final effect on the plants.

Response 3. We discussed the effects of nitrogen-fixing bacteria on plants in the section of DISCUSSION. “It has been seen that adding nitrogen changes the composition and activity of nitro-gen-fixing bacteria [53]. This has an effect on how seed production changes from year to year. Further investigation is warranted to empirically examine the potential impacts of this phenomenon”.

Bizjak, T.; Sellstedt, A.; Gratz, R.; Nordin, A. Presence and activity of nitrogen-fixing bacteria in Scots pine needles in a boreal forest: a nitrogen-addition experiment. Tree Physiol. 2023, tpad048.

  • The numbers in chemical formulas should be subscripted. For example, Lines 155 and 157.

Response 4. We replaced with “urea” by the reviewer 2.

  • Line 102 - first mention of S. nigrescens must be written in full - Saussurea nigrescens. 

Response 5. We introduced this species in first two sentences. We gave the full name of Saussurea nigrescens.

  • Lines 102-103 - The field experiment was conducted utilizing the nigrescens in Tibetan meadow. Explain why you selected this plant model. 

Response 6. Saussurea nigrescens demonstrates self-incompatibility and serves as a dominant herbaceous species in alpine meadows. Our previous study showed that the abundance of honey bees evolved to reduce nectar production. Recently, we observed that the production of its seeds is adversely affected by various environmental factors.

We demonstrated this in the section of INTRODUCTION.

  • Line 118 - with an altitude of 3500 m. Explain why you studied this place.

Response 7. Saussurea nigrescens frequently exhibits growth patterns within altitudinal ranges spanning from 2900 to 4300 m. In this study, the effects of nitrogen addition on interannual variations in seed output was carried out within the permanent plots of the Hongyuan Alpine Meadow Ecosystem Research Station, and the altitude is 3500 m.

  • Line 396 - absence of nectar-poor flowers - it should be probably - absence of nectar-rich flowers.

Response 8. We replaced with it.

  • Explain the use as pollinator honey bees Apis mellifera.

Response 9. Apiculture has been consistently practiced in the alpine meadows of the Zoige Plateau, located in the eastern region of the Tibetan Plateau, since the year 1981. This technique involves the utilization of the honey bee species known as Apis mellifera. From late July to early August, honey bees engaged in foraging activities exhibit a substantial reliance on S. nigrescens as a primary source of nectar collection. It is worth noting that the re-productive success of S. nigrescens is significantly influenced by the presence of A. mellifera.

We added the information in main text (Lines 152-158)

  • How yak grazing changed the grassland?

Response 10. The grassland being studied has been subjected to yak grazing for a period of less than twenty years, specifically occurring during the winter seasons (from September to May).

  • Lines 373-374 = The extent to which the dose of nitrogen reduces nectar production remains unknown. - It is already known - Cite this article - Vaudo, A.D., Erickson, E., Patch, H.M. et al. Impacts of soil nutrition on floral traits, pollinator attraction, and fitness in cucumbers (Cucumis sativus L.). Sci Rep 12, 21802 (2022). https://doi.org/10.1038/s41598-022-26164-4

Response 11. Here, we want to emphasize the effects of nitrogen levels on S. nigrescens remains unknown. We rewrote this sentence as “The extent to which the dose of nitrogen reduces nectar production in S. nigrescens remains unknown”.

  • English language and style are fine and minor spell check required.

Response 12. We checked the misspelling.

Reviewer 2 Report

This paper contains many data and results from 3 years field research. Analysis by "generalized linear mixed models" are fine. I agree this paper will be published in this journal. I add some comments directrly on the manuscrpt. Most of them are technical.  

Author Response

Reviewer 2

L 99-101 Addition of nitrogen to natural vegetation is good or not recommended? If possible, discuss at discussion about this (You might already mention in the text...).

Response 1. The augmentation of nitrogen concentrations has a positive impact on the seed production of S. nigrescens, especially when combined with a dosage of 8 g·m-2 of nitrogen.

We added the information in last paragraph.

L 102 Explain more about this plant, why you selected or focused on this plant.

Response 2. 

Saussurea nigrescens demonstrates self-incompatibility and serves as a dominant herbaceous species in alpine meadows. Our previous study showed that the abundance of honey bees evolved to reduce nectar production. Recently, we observed that the production of its seeds is adversely affected by various environmental factors. We demonstrated this in the section of INTRODUCTION.

L155 Better to change "urea". (NH2)2CO is better for urea. Why you select urea? Should be explained.

Response 3. We relaced with urea. The utilization of ammonium nitrate is recommended for conducting nitrogen deposition tests. However, ammonium nitrate is presently classified as a regulated substance, therefore limiting its use in China. Consequently, the only viable alternative is to substitute it with urea.

L 248 I think it is better to change these S-Figures and Tables are better to change to normal Figure and Tables to explain the results for readers. 

Response 4. We removed all the S-Figures and S-Tables in the main text.

L 266 Figure 2 comes first (before Figure 1). It is better to change these Figures. As for Figure 2, it is better to enlarge for the readers to read letters for explanation (abbreviations).

Response 5. We exchanged.

L 294-295 Better to add this explanation on Figure (a) instead of "sup". Better to add (b) instead of "re".

Response 6. We added.

L 297-300 Explanation of Box plot. Box plot are well-known, I think no need to explain here. But OK if prefer.

Response 7. We deleted.

L 301 P -> p (small)?

Response 8. We changed.

L 312 Better to explain in the Figure again.

Response 9. We added.
